# TRIMming Type I Interferon-Mediated Innate Immune Response in Antiviral and Antitumor Defense

**DOI:** 10.3390/v13020279

**Published:** 2021-02-11

**Authors:** Ling Wang, Shunbin Ning

**Affiliations:** 1Department of Internal Medicine, Quillen College of Medicine, East Tennessee State University, Johnson City, TN 37614, USA; wangl3@etsu.edu; 2Center of Excellence for Inflammation, Infectious Diseases and Immunity, Quillen College of Medicine, East Tennessee State University, Johnson City, TN 37614, USA

**Keywords:** TRIMs, ubiquitination, PRR, IFN-I, IRFs

## Abstract

The tripartite motif (TRIM) family comprises at least 80 members in humans, with most having ubiquitin or SUMO E3 ligase activity conferred by their N-terminal RING domain. TRIMs regulate a wide range of processes in ubiquitination- or sumoylation-dependent manners in most cases, and fewer as adaptors. Their roles in the regulation of viral infections, autophagy, cell cycle progression, DNA damage and other stress responses, and carcinogenesis are being increasingly appreciated, and their E3 ligase activities are attractive targets for developing specific immunotherapeutic strategies for immune diseases and cancers. Given their importance in antiviral immune response, viruses have evolved sophisticated immune escape strategies to subvert TRIM-mediated mechanisms. In this review, we focus on their regulation of IFN-I-mediated innate immune response, which plays key roles in antiviral and antitumor defense.

## 1. Introduction

In mammalians, interferons (IFNs) include three types, type I, II, and III. Type I IFNs (IFN-Is) include the majority of 26 isoforms of IFNα that are encoded by 13 genes, and one IFNβ that is encoded by the single gene IFNB, as well as other minor subtypes, including IFNε, IFNκ, IFNω, IFNδ, IFNτ, and IFNζ. IFNαs are mainly secreted by plasmacytoid dendritic cells (pDCs) and IFNβ is mainly secreted by fibroblasts. All IFN-Is signal through the integral membrane IFNAR1 and -2 heterodimer, and play crucial roles in the first line of innate immune response and subsequent adaptive immune response in response to viral or bacterial infections [1].

Importantly, recent studies have shown that IFN-Is play a dual role in chronic viral infections. At the early stage of infection, they have potent antiviral activity. However, at late stages, a low level of prolonged IFN-I signaling, exemplified by chronic infection of viruses such as HIV and HCV [2,3,4], triggers long-term chronic immune activation that proceeds to T cell exhaustion and inflammaging/immunosenescence in both direct and indirect manners [5,6] and therefore serves as a bridge that links innate and adaptive immune responses [4,5,7,8,9,10]. For example, the engagement of TLR7 in HIV-infected CD4^+^ T cells induces anergy/unresponsiveness, accounting for the impaired T cell function by chronic HIV infection [11]. A prolonged IFN-I response also facilitates the establishment of TME (tumor microenvironment) [4,5,7,12,13,14]. IFN-Is also play crucial roles in cellular development and homeostasis [5,6,15,16,17]. Aberrant production of IFN-Is is associated with many types of diseases, including autoimmune disorders and cancers [6,18,19,20]. Therefore, it is of fundamental importance to understand the precise mechanisms of how IFN-Is are regulated in different biological contexts [21,22].

Ubiquitination is a pervasive theme equally important to phosphorylation of proteins in myriad processes. Ubiquitin (Ub) is a 76-amino acid protein that is ubiquitously distributed and highly conserved throughout eukaryotic organisms. The Ub protein can be free or conjugated to a lysine site of a protein substrate through its 3’-end. This conjugation process involves E1 activating enzyme, E2 conjugating enzyme, and E3 ligase, with the E3 ligase determining the specificity of the substrate. Ub itself has seven internal lysine residues (K6, K11, K27, K29, K33, K48, and K63), and each can serve as the Ub target to link another Ub. If only a single Ub is conjugated to each lysine site of the substrates, it is called mono (also only one lysine site on the substrate) or multi (more than one lysine site on the substrate) ubiquitination. If the substrate is Ub itself, polyubiquitin chains will be formed on the substrate. Usually, a polyubiquitin chain contains more than 4 Ub molecules. In the last decade, non-canonical ubiquitination types on serine, threonine, and cysteine sites other than lysine site have been identified, and their importance in specific cellular functions has been recognized [23,24].

The most well-understood type of ubiquitination is K48-linked polyubiquitination, which is principally known as the major process whereby proteins are targeted for proteasomal degradation through the 26S proteasome. Later, nonproteolytic types of polyubiquitination (represented by K63-linked polyubiquitination), monoubiquitination, and linear ubiquitination have been gradually identified [25,26,27]. More recently, other ubiquitination-like modifications (e.g., sumoylation, acetylation, ISGylation, neddylation, palmitoylation, and UFMylation) have also been discovered. The roles of these posttranslational modifications (PTMs) in a myriad of cellular processes, such as receptor internalization (endocytosis), vesicle trafficking, immune response and inflammation, DNA damage response, autophagy, and cell death, have been greatly appreciated [27,28,29,30,31,32,33,34,35].

IFN-I production is controlled at multiple layers to ensure appropriate mounting of antiviral and antitumor immune responses. It is clear that both host and viral ubiquitin systems play pivotal roles in IFN-I-mediated innate immunity and in cellular transformation mediated by oncogenic viruses represented by EBV (Epstein-Barr Virus), KSHV (Kaposi’s sarcoma-associated herpesvirus), and HPV (human papillomavirus) [36,37,38,39,40,41]. This review focuses on IFN-Is, and we summarize the current research on the TRIM family in modulating IFN-I-mediated innate immune response in antiviral and antitumor defense.

## 2. PRR Signaling Pathways to IFN-I Production

IFN-Is are produced downstream of the signaling pathways of host germline-encoded pathogen recognition receptors (PRRs), which are expressed on the cell membrane or in the cytoplasm of the cells of the innate immune system, in response to pathogen-associated molecular patterns (PAMPs) that include pathogenic nucleic acids, LPS, and proteins, or in response to host damage-associated molecular patterns (DAMPs), such as self-nuclei acids, heat-shock proteins, and HMGB1. Recognition of PAMPs or DAMPs by PRRs triggers signal cascades that activate the transcription factors, including NFκB, Interferon regulatory factors (IRFs), and AP1, or activate caspase-mediated cell death and inflammation.

PRRs include the well-known transmembrane Toll-like receptors (TLRs) (Figure 1) and an increasing pool of “Toll-free” receptors [22]. Endosomal TLRs (TLR3, -7, -9, and murine TLR8) and endocytic TLR4, as well as cytoplasmic RIG-I and cGAS, amongst others, are able to recognize pathogenic or host cell nucleic acids (LPS for TLR4) to activate IRFs in addition to NFκB and AP1, which induce IFN-Is and also pro-inflammatory cytokines [42]. Self-nucleic acids are derived from the nucleus or mitochondria of the cells suffering from endogenous or exogenous stresses, such as DNA replication, oxidative stress, DNA damage, and cell death [43,44,45,46,47]. While the transcription of IFNαs is solely dependent on IRFs, full transactivation of the IFNβ promoter requires the cooperation of IRFs, NFκB, and other co-factors in the transcriptional complex named enhanceosome [48].

## 3. The TRIM Family

The tripartite motif (TRIM) family of proteins is large and includes at least 80 members in humans, with most having E3 ligase activity for target-specific ubiquitination, and plays crucial roles in innate immunity, transcription, autophagy, and carcinogenesis [49,50]. The N-terminal TRIM motif includes the conserved RBCC domain that comprises of three subdomains: 1 RING domain that confers with E3 ligase activity (8 human TRIMs do not have the RING domain), 0~2 B-box ZNF domains (B1+B2 or B2 alone), and 0~1 coil–coil region that is associated with B-boxes. According to the diversity of the C-terminuses and genomic organization, TRIM proteins are grouped into Group1 and Group 2. Members in Group 1 possess a variety of C-terminal domains (COS, FN3, ACID, PRY, PHD-BROMO, FIL, NHL, MATH, ARF, and TM) and exist in both vertebrate and invertebrates, and those in Group 2 possess a C-terminal SPRY domain, and they are absent in invertebrates (Figure 2) [49,50,51]. The SPY-SPRY domain is critical for TRIM proteins’ interaction with their substrates.

Many TRIMs are inducible by IFN-Is, and play crucial roles in IFN-I-mediated innate immune regulation, with the involvement of ubiquitination and sumoylation in most cases [50,52,53,54,55,56,57,58]. These TRIMs can target most if not all components of the PRR and Jak-STAT1 IFN-I pathways, including different ligands (PAMPs and DAMPs); the receptors such as TLRs, cGAS, and DDX41; the adaptors MyD88, TRIF, STING, and TRAF6 and -3; the kinases IKKs and TAK1; and the transcription factors IRF3 and -7 and NFκB (Table 1). TRIM genes evolve parallelly with the immune system, further supporting their roles as regulators of immune responses [51,53].

## 4. TRIMs in Regulating PRR Signaling Pathways to IFN-I Production

Upon binding to PAMPs or DAMPs, PRRs trigger signals that transmit via unique adaptors to the Ub E3 ligase TRAF6 or -3 and then orchestrate to activate the kinase cascades IKKs, IRAKs, and MAPKs for the activation of the transcription factors NFκB, IRFs, and AP1. Ubiquitination regulates the cellular trafficking, stability, complex assembly, and activity of different components in PRR signaling cascades (Figure 3).

Viral PAMPs and other viral components can be targeted by the host Ub system, including a subset of TRIMs, for ubiquitination-mediated degradation in most cases, and thus the IFN-I response is blocked at the very beginning to suppress viral replication, with some examples listed in Table 1 [54,58]. Of note, TRIM5α (also TRIM22) targets HIV1 Gag and plays a unique role in restricting HIV1 infection (and other retroviruses), implicating a potential clinical application [61,62]. In fewer cases, TRIMs can promote viral entry and replication by targeting viral proteins. For example, TRIM7 targets Zika virus envelope protein E for K63-linked ubiquitination that enhances viral attachment to the cell surface and promotes viral entry [66]. VP35, the Ebola virus polymerase co-factor, has IFN-I inhibitory activity. TRIM6 promotes VP35 polyubiquitination to enhance viral infection [64].

### 4.1. TRIMs in Regulating the PRRs

TLR3 senses dsRNA and TLR9 senses CpG DNA, whereas TLR7 and TLR8 sense ssRNA, in endosomes, leading to IFN-I production. TRIM3 promotes K63-linked ubiquitination of TLR3 at K831 (human), facilitating ESCRT (endosomal sorting complex required for transport)-mediated TLR3 sorting to endosomes, where TLR3 activates downstream signaling [59]. TRIM7-mediated ubiquitination promotes TLR4 activation of NFκB, AP1, and IRF3 [67].

Cytoplasmic ssRNA and dsRNA are mainly recognized by RIG-I and MDA5 respectively. RIG-I activation is promoted by K63-linked ubiquitination mediated by TRIM4 [60] or TRIM25 [121,122], and MDA5 activation is promoted by K63-linked ubiquitination mediated by TRIM65 [164]. However, the biological role of TRIM25 in the activation of RIG-I has been challenged by the in vivo evidence that the deletion of the gene encoding TRIM25 did not impair, but the deletion of that of RIPLET/RNF135 completely abrogated, RIG-I-mediated IFN-I response [165]. TRIM38 stabilizes RIG-I and MDA5 by sumoylation [146]. In contrast, TRIM40 promotes RIG-I and MDA5 proteasomal degradation through both K27- and K48-mediated ubiquitination forms [153]. The membrane-anchored TRIM13 and TRIM59 also promote RIG-I activity but negatively regulate MDA5-mediated IFN-I responses, through unclear mechanisms that may involve additional cofactors [79].

Cytosolic DNA, which is derived from self DNA or from invading pathogens, is mainly sensed by the cGAS-STING pathway and additionally by DDX41, DAI, and MRE11, amongst others [22]. The DNA fragment bound to cGAS is processed by cGAS synthetase activity to generate the second messenger cyclic GMP-AMP (cGAMP) that then binds to STING. cGAS stability and synthetase activity are balanced by ubiquitination, which is mediated by an increasing pool of E3 ligases including TRAF6, RNF185, and several TRIMs [166]. K48-linked ubiquitination of cGAS at K414 is targeted for degradation by p62-mediated selective autophagy [167]. TRIM14, a mitochondrial protein that lacks the RING domain, recruits USP14 to cleave K48 chains on cGAS to stabilize cGAS [85]. TRIM38 also stabilizes cGAS, however, through promoting cGAS sumoylation at early but K48 ubiquitination at the late stage of signaling at the same site K479 (K464 in mice) [147]. TRIM41 and TRIM56 both promote cGAS activation through mono-ubiquitination [155,159]. TRIM21 (known as Ro52), an autoantigen in patients with SLE (systemic lupus erythematosus), targets DDX41 for degradation [97].

### 4.2. TRIMs in Regulating the Adaptors

TLRs transmit signals via the adaptors MyD88 or TRIF. MyD88 is required for all TLRs except for TLR3, whereas TRIF is involved in both TLR3 and TLR4 pathways to the activation of IRFs. A subset of TRIMs regulate TRIF activity and stability (Table 1, Figure 3), but to our knowledge, so far, no report shows that MyD88 is targeted by any TRIM, although other E3 ligases target MyD88 for activation or degradation [168]. TRIM56 and TRIM62 promote TRIF activation [86,161], but TRIM32 and TRIM38 target TRIF for degradation. Mechanistically, TRIM56 physically interacts with TRIF upon viral infection, and its E3 ligase activity is not required for activation of TRIF [161]. TRIM32 also interacts with TRIF but targets TRIF to TAX1BP1-mediated selective autophagy for degradation [142]; whereas TRIM38 targets TRIF K228 for K48 ubiquitination, leading to proteosome-mediated degradation [150,151]. TRIM8 promotes K6- and K33-linked ubiquitination of TRIF and therefore disrupts TRIF-TBK1 interaction to specifically inhibit TLR3/4-mediated immune response [68].

In the cytosolic cGAS-STING DNA sensing pathway, TRIM56 promotes STING (also known as MITA) activation through K63-linked ubiquitination [160], in addition to its ability to promote cGAS activation through mono-ubiquitination [159]. TRIM32 also promotes STING activation through K63-linked ubiquitination [141]. In addition to K63-linked polyubiquitination, STING activation is also mediated by other atypical ubiquitin chains such as AMFR (known as RNF45)-mediated K27 ubiquitination [169]. The deubiquitinase USP21 can negatively regulate STING activity by removing K27- and K63-linked ubiquitin chains of STING [170]. TRIM38 stabilizes both STING and cGAS [147]. However, TRIM29 and TRIM30α promote STING degradation through K48-linked ubiquitination [133,134,137].

In the cytosolic RIG-I RNA sensing pathway, TRIM14 binds to the downstream adaptor MAVS on mitochondria. Upon viral infection, TRIM14 undergoes K63-linked self ubiquitination at K365, consequently recruiting NEMO to MAVS signalosome for NFκB activation [84]. TRIM25 and TRIM29 promote MAVS degradation by K48- and K11-linked ubiquitination, respectively [123,135]. However, TRIM21/Ro52 triggers K27-linked ubiquitination [98], and TRIM31 triggers K63-linked ubiquitination [139], of MAVS, and TRIM44 stabilizes MAVS by preventing its ubiquitination and degradation [156], to promote RIG-I signaling pathway. TRIM21/Ro52 also negatively regulates RIG-I-mediated antiviral immunity through its interaction with FADD, promoting ubiquitination-mediated IRF7 degradation [99]. In addition, TRIM15 promotes both RIG-I- and MDA-mediated antiviral activity, likely at the level or upstream of MAVS [86].

### 4.3. TRIMs in Regulating TRAF3/6

TRAF3 is critical for activation of IRFs downstream of PRRs leading to IFN-I production [171,172], whereas TRAF6 is generally required for NFκB and AP1 activation in these settings. Both TRAF3 and TRAF6 are members of the TRAF E3 ligase family, and they can catalyze self-ubiquitination.

TRAF3 activity and stability are regulated by ubiquitination. Its activation requires K63 ubiquitin conjugation, which is mediated by TRIM24, TRIM35, TRAF3 itself, TRAF6, and other E3 ligases in distinct contexts [117,144,173,174]. TRIM23 also interacts with TRAF3 in coimmunoprecipitation assay, but the function of TRIM23 in this setting is unknown [114].

As such, TRAF6 activation is promoted by different ubiquitination forms mediated by TRAF6 itself, mouse TRIM12c (the homolog of human TRIM5α) [62], and TRIM13 [80]. Furthermore, TRIM23 interacts with TRAF6 and CMV UL144 in a complex, which promotes TRAF6 autoubiquitination and downstream NFκB activation independently of TRIM23 E3 ligase activity [115]. TRIM38, however, promotes TRAF6 K48-linked ubiquitination and degradation downstream of TLR signaling in human macrophages [148] but not in mouse cells [150].

### 4.4. TRIMs in Regulating the Kinase Cascades for Activation of IRFs and NFκB

For IFN-I transcription, the IKK family members TBK1, IKKε, and IKKα (IKKβ for IRF5) are responsible for the activation of IRFs, with the involvement of other kinases including IRAKs, RIP1, and the MAPK family member TAK1. The IKK family members IKKα, IKKβ, and NEMO (IKKi), however, are for NFκB activation. A pool of kinase-interacting partners and adaptors are also involved, such as TANK, NAP1, TAB1/2, FADD, and TRADD. Ubiquitination is involved in the modulation of the kinase cascades.

TBK1 plays a critical role in the activation of IRF3 downstream of various PRR pathways. TRIM23 has both E3 ligase and GTPase activities. K27-linked autoubiquitination of TRIM23 is essential for its GTPase activity, which then facilitates TBK1 activation to phosphorylate p62, promoting p62-mediated selective autophagy [116]. p62-mediated selective autophagy is known to promote the degradation of multiple components in PRR pathways (Table 1). As such, K27-linked autoubiquitination of TRIM26 facilitates the recruitment of NEMO to the constitutive TBK1-TRIM26 complex, promoting TBK1 recruitment to VISA signalosome and consequent activation [126]. TRIM9s interacts with TBK1. Viral infection promotes TRIM9s K63-linked autoubiquitination, recruiting GSK3β, a TBK1-interacting partner that promotes TBK1 phosphorylation [175], to the TRIM9s-TBK1 complex to promote TBK1 activation [75], whereas TRIM11 or TRIM14 interaction with TBK1 inhibits TBK1 activation in ubiquitination-independent manners [77,83]. TRIM27, however, promotes TBK1 K48-linked ubiquitination and degradation [128,129,130].

TRIM21/Ro52 catalyzes free K63 ubiquitin chains, which serve as a scaffold to promote TAK1 activation [100,101]; however, TRIM21/Ro52 promotes IKKβ autophagic degradation through mono-ubiquitination [102]. TRIM5α promotes TAK1 K63-linked ubiquitination and activation [63]. TRIM8 activates TAK1 by promoting its K63-linked ubiquitination [69,70]. TRIM27 interacts IKKα, -β, and -ε, in addition to TBK1, in the IKK family members and inhibits their activity and activation downstream of NFκB/IRFs, in which its E3 ligase activity is not required [131]. NEMO is activated by K27-linked ubiquitination mediated by TRIM23 [114] but is degraded by K48-linked ubiquitination mediated by TRIM13 or TRIM29 [81,136]. TRIM40 promotes NEMO neddylation to inhibit NFκB-mediated inflammation in gastrointestinal cancer [154].

TRIMs also indirectly modulate kinase activity by regulating kinase-interacting partners or adaptors. TRIM22, TRIM30α, and TRIM38 promote the TANK-binding partners TAB2/3 for degradation [113,138,150]. The TANK family protein NAP1 (NFκB activating kinase-associated protein 1), which is required for TRIF-mediated activation of IRFs, is targeted by TRIM38 for degradation through K48-linked ubiquitination [149]. β-TrCP is a component of the SCF E3 ubiquitin ligase complex, which promotes IκBα proteasomal degradation and the p52 precursor p100 processing for NFκB activation. The brain-specific TRIM9 interacts with β-TrCP and prevents its function, consequently inhibiting NFκB-mediated inflammation [75,76]. TRIM39 stabilizes the negative regulator of TLR signaling pathways, Cactin, and therefore inhibiting the activation of downstream NFκB and IRFs [152]. As such, TRIM59 interacts with the signal adaptor protein ECSIT (evolutionarily conserved signaling intermediate in Toll pathways) and negatively regulates the activation of NFκB and IRFs [162]. TFG (TRK-Fused Gene) is a potential component of the TRAF3-TBK1 signalosome downstream of PRR pathways [163,176]. TRIM68 promotes TFG lysosomal degradation and therefore inhibits TRIF-mediated IFNβ production [163].

### 4.5. TRIMs in Regulating the Transcription Factors IRFs and NF-κB for IFN-I Production

Accumulating evidence has shown that ubiquitination and ubiquitination-like modifications directly target the IFN-I transcription factors IRFs and NFκB for their activation in antiviral and antitumor immunity.

As a multifaceted regulator in IFN-I-mediated defense, TRIM21/Ro52 directly targets a subset of IRFs for degradation, and in turn, it is directly induced by IRF1 and -2 [177]. TRIM21/Ro52 acts as a double edge sword to regulate IRF3: TRIM21/Ro52 directly targets phosphorylated IRF3 (p-IRF3) for proteasomal degradation, which is enhanced by tyrosine phosphorylation of TRIM21 at Y393 [104,105]. TRIM21/Ro52 also interacts with multiple autophagy components, including the selective autophagy receptor p62, ULK1, and Beclin1, and serves as a platform to facilitate the assembly of the autophagy apparatus, which directs IRF3 for lysosomal degradation [106]. On the other hand, TRIM21 interferes with the interaction between p-IRF3 and the prolyl isomerase Pin1 independently of its E3 ligase activity to protect p-IRF3 from Pin1-mediated ubiquitination and degradation [103]. Furthermore, Pin1 itself interacts with sumoylated TRIM19/PML isoform IV (TRIM19IV), which results in its recruitment to PML-NBS, preventing p-IRF3 from degradation [91]. TRIM21 also promotes proteasomal degradation of the phosphorylated forms of IRF7 as well as V1 and V5 isoforms of IRF5 [107,108] but promotes IRF8 activation [109]. In addition to its direct targeting p-IRF7 for degradation, as mentioned above, TRIM21 interacts with FADD downstream of RIG-I signaling, promoting ubiquitination-mediated IRF7 degradation [99].

In addition to TRIM21, TRIM26 targets IRF3 [127], and TRIM35 targets IRF7 [145], for proteasomal degradation, most likely in the nuclear compartment (i.e., TRIM26 and -35 target phosphorylated forms of IRF3 and -7, respectively). Pin1 acts as a double edge sword to regulate IRF7 downstream of TLR7/9: Pin1 activates the IRF7 kinase IRAK1, leading to IRF7 phosphorylation and activation [178]. On the other hand, Pin1 mediates p-IRF7 for degradation [71]. Like TRIM21-mediated protection of p-IRF3 from Pin1-mediated degradation [103], TRIM8 protects p-IRF7 from Pin1-mediated degradation [71]. The transcriptional corepressor TRIM28 is a SUMO E3 ligase that promotes IRF7 sumoylation and negatively regulates its transcriptional activity [132]. 

TRIM19/PML, as a transcriptional repressor, suppresses NFκB-mediated gene transcription via its C-terminus independently of its SUMO E3 ligase activity and therefore inhibits TNFα-induced apoptosis in MEFs [93]. Controversially, a later report shows that TRIM19/PML promotes p65 phosphorylation and TNFα-induced NFκB activity in MEFs [94]. TRIM20 promotes NFκB activation in two ways: promoting p65 nuclear translocation and IκBα degradation [96]. TRIM20 also functions as a selective autophagy receptor targeting NLRP3 for autophagic degradation [106].

## 5. TRIMs in Regulating the Jak-STAT IFN-I Signaling

The level of initial IFN-Is produced downstream of PRR pathways upon viral infection is relatively low due to the low level of endogenous IRF7 protein; these priming IFN-Is then secret to outside of the cell in autocrine and paracrine manners, and bind to IFN-I receptor (IFNAR) on other cells, consequently triggering the Jak-STAT IFN-I pathway, which serves as the second phase of antiviral response by inducing the expression of more IRF7, which in turn participates in IFN-I production downstream of PRR signaling, therefore amplifying the IFN-I production in a positive regulatory circuit (Figure 4) [42]. 

Jak-STAT pathways are well known to be negatively regulated by two families: SOCS (Suppressor of cytokine signaling) and PIAS (Protein inhibitor of activated STAT). TRIM8 can shuttle between the cytoplasm and the nucleus [74], and has multiple functions to promote IFN-I signaling. TRIM8 promotes proteasomal degradation of SOCS1 and PIAS3 presumedly in the cytoplasm, and also nuclear TRIM8 promotes PIAS3 nucleus-cytoplasm translocation to inhibit PIAS3 activity [72,73,74]. SOCS1 not only inhibits Jak1 activity by directly binding to phosphorylated Jak1 in the IFN-I Jak-STAT signaling but also acts as a ubiquitin E3 ligase that targets phosphorylated IRF3 and IRF7 (both also targeted by SOCS3 that recruits the Cul-RBX2 E3 complex) for proteasomal degradation in the nucleus [179]. As such, PIAS3 acts as a SUMO E3 ligase that inhibits IRF1 transcriptional activity through sumoylation in addition to its ability to inhibit STAT3 [180]. TRIM14 negatively regulates IFN-I signaling in mouse macrophage in response to *Mycobacterium tuberculosis* infection by serving as a scaffold that bridges TBK1-STAT3 interaction promoting STAT3 S727 phosphorylation, consequently inducing SOCS3 expression that inhibits IFN-I signaling by targeting phosphorylated IRF3 and IRF7 as well as TBK1 for proteasomal degradation [83]. The nuclear protein TRIM19/PML promotes ISGF3-mediated gene expression by facilitating STAT1 gene transcription and STAT2 protein stabilization, as well as the accumulation of both activated STAT1 and -2 to chromosome [90].

IKKε is not only responsible for the activation of IRF7 and -3 but also plays a role in balancing IFN-I and IFN-II Jak-STAT signaling pathways in immune responses [181]. TRIM6 catalyzes free chains of K48, which promotes IKKε oligomerization and activation to facilitate STAT1 S708 phosphorylation and IFN-I signaling [65]. However, TRIM24 can inhibit retinoic acid-induced STAT1 transcription by interacting with the transcription factor RARα on the STAT1 gene promoter [118].

IFN-Is establish an antiviral state in both virus-infected cells and uninfected bystander cells, by inducing the expression of over 300 ISGs (IFN-stimulated genes) [6]. Many components of the PRR signaling pathways, such as RIG-I, cGAS, STING, IRF1, and IRF7 belong to ISGs. In addition to these components, many other ISGs, including some TRIMs themselves, are also directly regulated by TRIMs. For example, TRIM11 promotes TRIM5 turnover dependently on its RING domain [78]. Ubiquitination-like modifications, such as sumoylation and ISGylation, are involved in IFN-I-mediated defense mechanisms [30,34,182,183,184,185]. TRIM19/PML mediates global sumoylation [92], and TRIM25 functions as an ISG15 E3 ligase that mediates ISGylation [124]. Further, TRIM25 has been recently reported to be required for the stability of several ISG products [186]. The zinc-finger antiviral protein ZAP, as an ISG, is activated by TRIM25-mediated ubiquitination to inhibit viral genome translation [125]. The tumor suppressor p53 is also an ISG inducible by IFN-Is [187]. TRIM24 promotes p53 ubiquitination and degradation and, in turn, is inducible by p53 [119]. ATM phosphorylates TRIM24 at S768 and promotes its degradation, stabilizing p53 [188]. Numerous TRIMs, in addition to TRIM24, regulate p53 activity and stability in direct or indirect manners [189].

## 6. Viral Strategies to Subvert TRIM-Mediated IFN-I Regulatory Mechanisms

Given the overwhelming evidence that TRIMs play versatile roles in antiviral defense, viruses have developed sophisticated strategies to counteract these TRIM-mediated mechanisms. We list some strategies invoked by herpesviruses and HPV in their lytic and latent infections.

The herpesviral deubiquitinases, including EBV BPLF1, KSHV ORF64, and CMV UL48, but not HSV1 (herpes simplex virus 1) UL36, can interact with the scaffold protein 14-3-3 and remove ubiquitin chains from TRIM25 in the RIG-I signalosome, inactivating RIG-I signaling to facilitate their lytic infection [190,191,192]. Another HSV1 deubiquitinase, VP1-2, removes K63 Ub chains from STING [193], which can be conjugated by TRIM32 and -56 [141,160]. HTLV1 (human T-cell leukemia virus 1) Tax and HBV (hepatitis B virus) polymerases also negatively regulate STING K63-linked ubiquitination to facilitate their infection [194,195]. The genomic DNA of EBV and other DNA viruses induce TRIM29 expression to suppress STING-mediated IFN-I immune signaling during lytic infection [134].

HSV1 ICP0 exploits the deubiquitinase USP7/HAUSP for immune escape by promoting USP7 from the nucleus to the cytoplasm, where USP7 removes K63 Ub chains from TRAF6 and NEMO, and removes K48 chains from TRIM27, to attenuate PRR signaling, likely independently on its E3 ligase activity [128,196]. However, ICP0 E3 ligase activity directly targets TRIM27 for proteasomal degradation to regulate HSV1 infection [197]. In addition, ICP0 directly targets many components of the PRR pathways including IFI16, and IRF3, and -7 in the nucleus, and MyD88, for degradation [198,199]. Given the importance of USP7 in negatively regulating ubiquitination-mediated IFN-I response, other herpesviruses also exploit USP7 to dampen IFN-I response. KSHV vIRF4 interacts with USP7 to inhibit p53-mediated antiviral activity [200], whereas EBV EBNA1 disrupts p53-USP7 interaction, to promote their latent establishment [201].

Sumoylation promotes stability of IRF3, RIG-I, MDA5, cGAS, and STING. The SUMO protease SENP2 can reverse the sumoylation process, leading to their degradation [146,147,202,203]. LMP1 was reported to inhibit SENP2 activity in EBV latency [204]. Thus, LMP1 regulation of SENP2-mediated desumoylation may represent one of many strategies of EBV to balance IFN-I response mediated by these PRR pathways to benefit its infection and latency establishment. We have shown that LMP1 exploits the host ubiquitin system to regulate IRF7 activity, including the induction of linear ubiquitination mediated by LUBAC to repress IRF7 activity and to promote NFκB activation in EBV latency [205,206,207].

TRIM19/PML is another important target attacked by various viruses for immune escape [208,209]. As a ubiquitin E3 ligase, HSV1 ICP0 preferentially targets the SUMO-modified TRIM19/PML isoforms, leading to their degradation [210], among its many roles in HSV1 lytic and latent infection [211]. EBV EBNA1 and KSHV LANA2 disrupt PML bodies [212,213]. HCMV IE1 binds to PML in a complex with ISGF3 to impede IFN-I signaling [90]. In contrast, HPV requires PML nuclear bodies to establish infection [214]. Other examples for viral targeting PML include that EMCV 3C protease promotes PML degradation [215] and that the rabies virus phosphoprotein P and its truncated mutants cause the redistribution of PML-NBs into the cytoplasm [216]. Interestingly, TRIM19/PML is a cellular ROS (reactive oxygen species) sensor that possesses basal antioxidant properties but also drives ROS-mediated p53 activation; its depletion promotes ROS accumulation and NRF2-mediated antioxidant defense. Thus, TRIM19 plays a role in ROS-mediated p53 responses, including senescence, cell survival/proliferation, and metabolism [95]. The production of ROS and RNS (reactive nitrogen species) is one of cancer hallmarks and the most common complication of persistent viral infection [217,218,219,220,221]. ROS/RNS also cause endogenous DNA damage that can activate the cGAS-STING pathway, which plays critical roles in natural anti-tumor or pro-tumor immunity as well as in ageing [46,47,222,223,224]. Moreover, we and others have shown that, in chronic viral infections, ROS/RNS couple with p62-mediated selective autophagy machinery [225,226], which serves as an alternative DNA damage repair strategy in these settings that have impaired traditional DNA repair machinery.

HPV E6 and E7 are two oncogenic proteins critical for HPV-mediated oncogenic transformation. E6 can hijack the host ubiquitin E3 ligase E6AP to inactivate the tumor suppressor p53 by promoting its ubiquitination, whereas E7 hijacks the host cullin 2-Zer1 ubiquitin ligase complex for ubiquitination-mediated degradation of the tumor suppressor Rb. Interestingly, E6 exploits the host deubiquitinase USP15 to target TRIM25 for ubiquitination-mediated degradation, to escape RIG-I-mediated immune response [227]. With a high throughput screen strategy, E6 and E7 were found to interact with a pool of UPS-related proteins, including TRIM32 and -72 for E7 [228]. LUBAC-mediated linear ubiquitination of TRIM25 also promotes its degradation [229]. We have shown that EBV LMP1 exploits LUBAC-mediated linear ubiquitination, which inhibits LMP1-stimulated IRF7 activity in EBV latency [207]. This finding implies that targeting TRIM25 by LUBAC may represent another strategy for EBV to escape IFN-I immune response in its latency.

Many other viruses are also known to subvert TRIM-mediated regulation of IFN-I network. For example, the matrix protein of Nipah virus targets TRIM6 for degradation to inhibit IKKε-mediated IFN-I antiviral response [230]. Identification of virus-specific products in this process is necessary for improving our understanding of the host-virus interactions.

## 7. Perspectives

TRIMs have diverse roles in host antiviral and antitumor defense systems, with or without the involvement of ubiquitination and alike modifications. Note that the roles of a given TRIM protein in different PRR signal pathways are probably distinct or even opposite, depending on the stimuli and cell contexts. Even more, the same TRIM has opposite outcomes on the same target through different mechanisms. For example, TRIM21 directly targets phosphorylated IRF3 for proteasomal degradation and also protect p-IRF3 from Pin1-mediated ubiquitination and degradation independently of its E3 ligase activity. In the meantime, TRIM21 indirectly regulates IRF3 activity by targeting the signal intermediators, such as DDX41, MAVS, FADD, TAK1, and the autophagy machinery, upstream of PRR pathways. The ultimate outcomes are manifested by the converge of these regulation events.

Besides those TRIM members listed in Table 1, other members in this family may have potential roles in IFN-I regulation. It is important to identify the unique role and targets of each TRIM member, which can be accelerated with different high throughput screening strategies, such as targeted proteolysis, combined with the CRISPR-mediated knockout or knockin techniques [231]. It is also important to understand how TRIMs mediate the crosstalk between immunity and cancer via the regulation of IFN-I network. Of note, selective autophagy, especially that mediated by p62, is emerging to play a crucial role in this process. As shown in Table 1, selective autophagy targets key components of PRR pathways for degradation and is required for TBK1 activation [116]. A subset of TRIMs, including those IFN-I regulators TRIM5α, -11, -14, -20, -21, -23, -28, -31, -32, -38, -39, -59, and -65 listed in Table 1, amongst others, regulate autophagy [49,56,232]. Since the cGAS-STING pathway plays a crucial role in oncogenesis, by sensing self DNA fragments derived from oxidative, infection, and other stresses, it is plausible that these TRIMs connect the cGAS-STING-mediated innate immunity and carcinogenesis via the autophagy machinery, which plays fundamental roles in antiviral and antitumor defense and is a potential target for cancer immunotherapy [233,234,235,236,237,238,239,240].

The regulation of the IFN-I network by TRIMs mostly requires their E3 ligase activity that possesses high substrate specificity, or their activity is regulated by specific enzymes, providing an applausive opportunity for drug targeting for clinical applications. However, there are no drugs directly targeting the E3 ligase activity of TRIMs so far, although some drugs targeting proteasomes or the autophagy machinery have been developed for this purpose. More efforts are desired towards this objective.

## Figures and Tables

**Figure 1 viruses-13-00279-f001:**
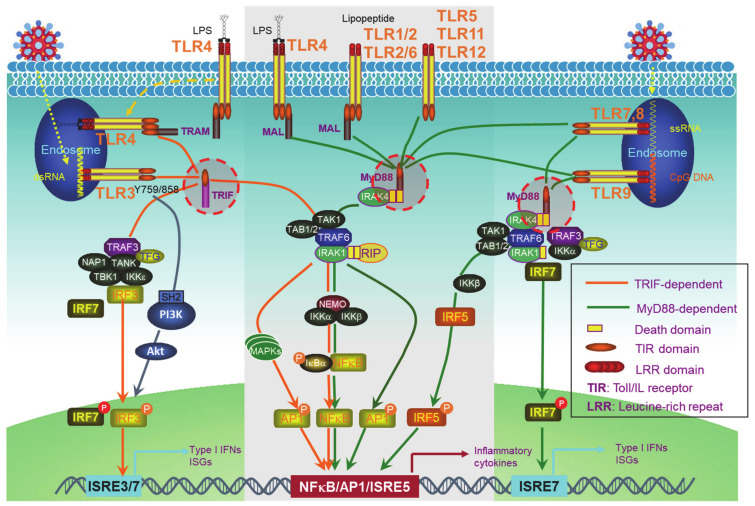
Toll-like receptor (TLR)signaling pathways. The TLR family has 13 members, among which, endosomal TLR3, -7, -9, and murine TLR8, and endocytic TLR4 are able to trigger signaling for IFN-I production. TLRs that cannot trigger the activation of Interferon regulatory factors (IRFs) (The middle part in the gray frame) do not contribute to IFN-I production. All the TLRs have a TIR domain in the cytoplasm, which recruits adaptor proteins also with a TIR domain at their C-terminus. TRIF and MyD88, two adaptor proteins, bridge all TLRs to downstream signaling molecules, leading to the activation of NFκB, IRFs, and AP1. IRF1, -3, -5, -7, and -8 are the transcription factors for both IFNα and IFNβ transcription in different cell contexts, but a full IFNβ transcription requires the enhanceosome complex that contains NFκB, IRF3, -7, ATF-2/c-Jun, and HMGIY (high mobility group I(Y)). ISRE: Interferon-stimulated response element. ISRE3/7: ISRE that binds to IRF3//7. ISRE7: ISRE that binds to IRF7.

**Figure 2 viruses-13-00279-f002:**
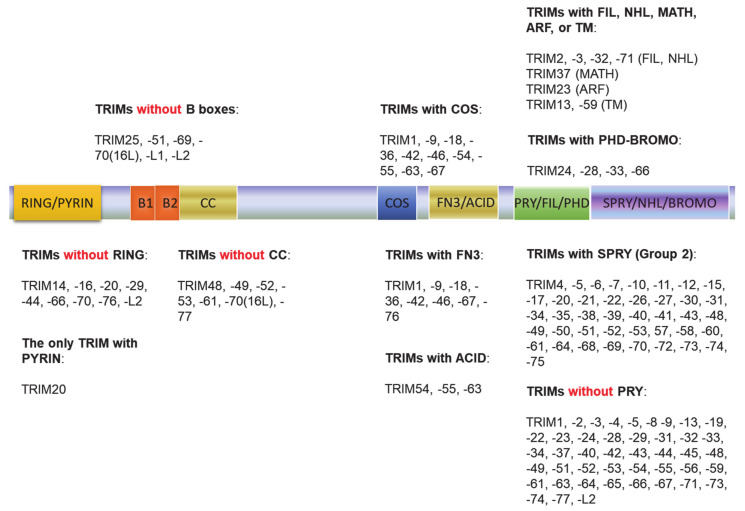
The tripartite motif (TRIM) family protein domain alignment. The TRIM family includes at least 80 members in humans. The N-terminal TRIM motif includes the conserved RBCC domain that comprises of three subdomains: 1 RING domain that confers with E3 ligase activity (8 TRIMs in humans do not have the RING domain), 0~2 B-box ZNF domains (B1+B2 or B2 alone), and 0~1 coil–coil region that is associated with B-boxes. According to the diversity of the C-terminuses and genomic organization, TRIM proteins are grouped into Group1 and Group 2. Members in Group 1 possess a variety of C-terminal domains (COS, FN3, ACID, PRY, PHD-BROMO, FIL, NHL, MATH, ARF, and TM) and exist in both vertebrate and invertebrates, and those in Group 2 possess a C-terminal SPRY domain, and they are absent in invertebrates.

**Figure 3 viruses-13-00279-f003:**
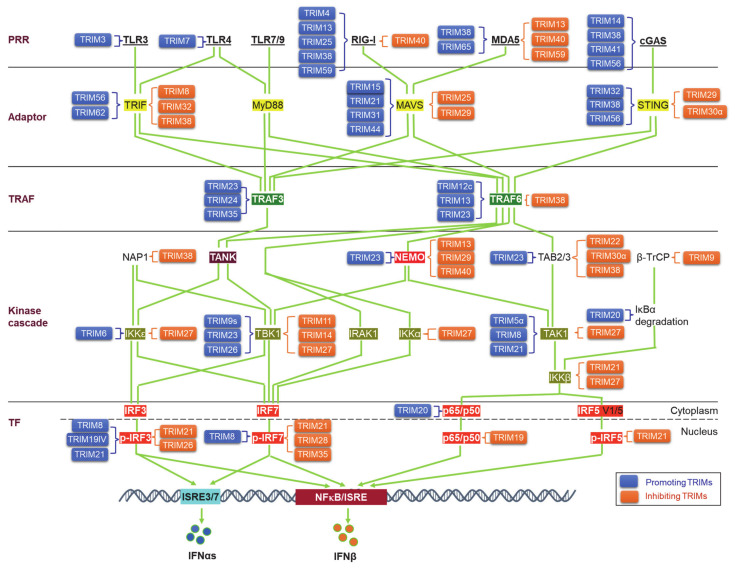
TRIM regulation of pathogen recognition receptor (PRR) pathways. TRIMs are involved in the regulation of the stability and activity of PRR components in all cascades of the signaling pathways, including ligands, receptors, adaptors, TRAFs, kinases and associated regulators, and the final transcription factors (TFs). An increasing pool of regulatory factors of the PRR pathways is also regulated by TRIMs (not shown). TRIMs promoting the stability or activity of the targets are shown on the left of the targets (blue), and those inhibiting the targets are shown on the right of the targets (brown). TRIM19IV and TRIM21 positively regulate phosphorylated IRF3 in indirect manners via Pin1. As such, TRIM8 positively regulates phosphorylated IRF7 in an indirect manner via Pin1. Other indirect regulations of these PRR pathways by TRIMs are not shown. TF: Transcription factor.

**Figure 4 viruses-13-00279-f004:**
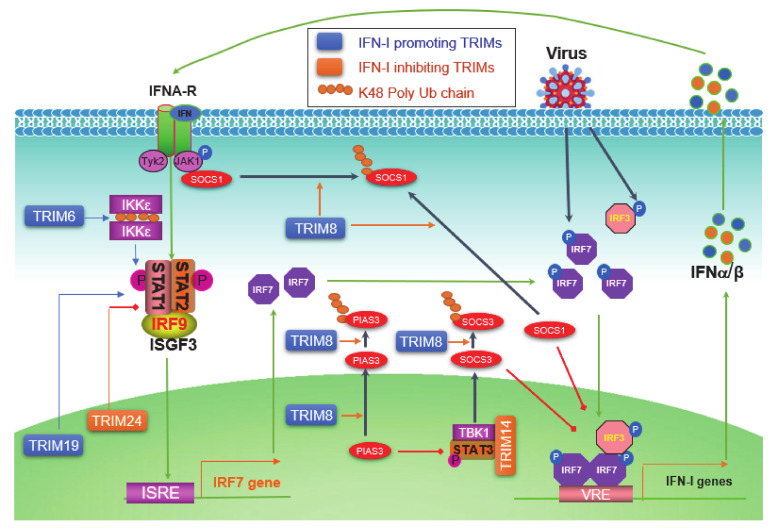
TRIM regulation of the Jak-STAT IFN-I signaling circuit. pathogen-associated molecular patterns (PAMPs) from pathogens activate PRR pathways, leading to the phosphorylation of constitutive high level of IRF3 and low level of IRF7, which induce a low level of IFN-Is. IFN-Is secrete from the cell and bind to IFN-I receptors (IFNAR1 and -2) on the cell membrane, followed by the recruitment and activation of Jak1 and Tyk2, leading to the phosphorylation and activation of STAT1 and -2. Phosphorylated STAT1 and -2 bind to IRF9 to form the ISGF3 (IFN-stimulated gene factor) complex, which functions as the transcriptional activator of more than 300 IFN-inducible genes (ISGs), including IRF7 itself. The induced IRF7 proteins then return to be activated by PRR pathways, and therefore constitute a positive regulatory circuit between IRF7 and IFN-Is, ensuring a potent production of IFN-Is to fight the invading pathogen. TRIM6 promotes free K48-linked Ub chains that serve as a platform to facilitate IKKε dimerization and activation. TRIM8 regulates the Jak-STAT IFN-I signaling at multiple points. The nuclear TRIM19 potentiates the transcription and activation of STAT1 and -2, and nuclear TRIM24 inhibits RARα-mediated STAT1 promoter activation. ISRE: Interferon-stimulated response element; VRE: Virus-responsive element.

**Table 1 viruses-13-00279-t001:** TRIMs in the regulation of Interferon (IFN)-I-mediated innate immune network.

TRIM	Synonym	Targets in the IFN-I Network	Ub Conjugation Type	Outcome of Conjugation	Selected References
TRIM3	RNF97	TLR3	K63	Promotes ESCRT-mediated TLR3 sorting to endosomes	[59]
TRIM4	RNF87	RIG-I	K63	Activation	[60]
TRIM5α	RNF88	HIV Gag	K48	Degradation	[61,62]
		TAK1	K63	Activation	[63]
	TRIM12c in mice	TRAF6	K63 (?)	Activation	[62]
TRIM6	RNF89	Ebola VP35	Poly	Promotes VP35 IFN-I inhibitory activity	[64]
		IKKε	Free K48	Activation of IKKε, leading to STAT1 activation	[65]
TRIM7	RNF90 GNIP	Zika virus envelope (E)	K63	Enhances virus attachment and entry into the cell	[66]
		TLR4	NA*	Promotes TLR4 activation	[67]
TRIM8	RNF27 GERP	TRIF	K6, K33	Disrupts the TRIF-TBK1 complex	[68]
		TAK1	K63	Activation	[69,70]
		IRF7		Protects p-IRF7 from Pin1-mediated proteasomal degradation in the nucleus	[71]
		SOCS1	K48 (?)	Degradation	[72]
		PIAS3	K48	Degradation	[73]
			Interaction (?)	Promotes PIAS3 nucleus-to-cytoplasm translocation	[74]
TRIM9s	RNF91 SPRING	TBK1	Interaction	Recruits GSK3β and TBK1, leading to TBK1 activation	[75]
TRIM9		β-TrCP	Interaction	Stabilizes IκBα	[75,76]
TRIM11	RNF92 BIA1	TBK1	Interaction	Inhibits TBK1 activation	[77]
		TRIM5	NA*	Degradation	[78]
TRIM13	RNF77 RFP2 CAR LEU5 DLEU5	RIG-I	Interaction	Potentiates RIG-I activity	[79]
		MDA5	Interaction	Inhibition	[79]
		TRAF6	K29	Activation	[80]
		NEMO	K48	Degradation	[81]
TRIM14	KIAA0129	HCV NS5A	K48 (?)	Degradation	[82]
		cGAS, TBK1	Interaction	Inhibition of autophagic degradation of cGAS	[83,84,85]
		MAVS	Interaction	Recruitment of NEMO to MAVS signalosome	[84]
TRIM15	RNF93 ZNF178 ZNFB7	MAVS	NA*	Promotes RIG-I-mediated IFN production	[86]
TRIM19	RNF71 PML MYL	HIV genome		Sequestrates HIV genome in the cytoplasm, blocking HIV transduction	[87]
		HFV Tas		Represses HFV transcription by preventing Tas binding to viral DNA	[88]
		LCMV Z		Inhibits LCMV replication	[89]
		hCMV IE1	Interaction	IE1 forms a complex with TRIM19-STAT1/2 to impede IFN-I signaling	[90]
		STAT1/2		Induction and stabilization, promoting IFN-I signaling	[90]
		Pin1 (by TRIM19IV)		Regulates the cellular distribution of Pin1	[91]
		Ubc9 (The only SUMO E2)		Required for IFN-induced global sumoylation	[92]
		NFκB		Inhibits NFκB-mediated transcription and survival	[93]
				Promotes IKKε-mediated p65 phosphorylation and NFκB activity	[94]
		ROS		Functions as an ROS sensor promoting p53 activation	[95]
TRIM20	Pyrin MEFV	p65	Interaction	Promotes p65 nuclear translocation	[96]
		IκBα		Promotes IκBα degradation	[96]
TRIM21	RNF81 Ro52 SSA1	DDX41	K48	Degradation	[97]
		MAVS	K27	Activation	[98]
		FADD	Interaction	Promotes IRF7 ubiquitination-mediated degradation	[99]
		TAK1	Free K63	Activates TAK1, leading to the activation of NFκB, AP1, and IRFs	[100,101]
		IKKβ	Mono-Ub	Autophagic degradation	[102]
		IRF3	Interaction	Protects p-IRF3 from Pin1-mediated proteasomal degradation	[103]
			K48	Targets IRF3 for proteosomal degradation	[104,105]
		Interacts with ULK1, Beclin1, and p62		Targets IRF3 for autophagic degradation	[106]
		IRF5	Various	Degradation of isoforms V1 and V5, but not V2 or V3	[107]
		IRF7	K48	Degradation	[108]
		IRF8	NA*	Activation	[109]
TRIM22	RNF94 STAF50	HIV Gag, LTR		Degradation	[110]
		Influenza A Virus NP		Degradation	[111]
		HCV NS5A	K48 (?)	Degradation	[112]
		TAB2	K48 (?)	Degradation	[113]
TRIM23	RNF46 ARD1 ARFD1	TRAF3	Interaction	Function not clear, likely promoting TRAF3-mediated antiviral activity	[114]
		TRAF6	Interaction	Activation of NFκB mediated by HCMV UL144	[115]
		NEMO	K27	Activation	[114]
		TBK1	K27 of TRIM23 (self)	Recruits and activates TBK1, inducing TBK1-mediated autophagy	[116]
TRIM24	RNF82 TIF1A	TRAF3	K63	Activation	[117]
		RARα	Interaction	Inhibits RARα activity and retinoic acid-induced STAT1 expression	[118]
		p53	K48 (?)	Promotes p53 ubiquitination and degradation	[119]
TRIM25	RNF147 ZNF147	Influenza virus vRNP		Blocks vRNA chain elongation	[120]
		RIG-I	K63	Activation	[121,122]
		MAVS	K48	Degradation	[123]
		ISG15		Functions as an ISG15 E3 ligase	[124]
		ZAP	K48, K63	Critical for ZAP inhibition of viral genome translation	[125]
TRIM26	RNF95 ZNF173 AFP	TBK1	K27 of TRIM26 (self)	Bridges TBK1-NEMO interaction, leading to TBK1 activation	[126]
		IRF3	K48	Degradation	[127]
TRIM27	RNF76 RFP	TBK1	K48	Degradation	[128,129,130]
		IKKα, IKKβ, IKKε	Interaction	Inhibition	[131]
TRIM28	RNF96 KAP1	IRF7	Sumoylation	Inhibition	[132]
TRIM29	ATDC	STING	K48	Degradation	[133,134]
		MAVS	K11	Degradation	[135]
		NEMO	K48	Degradation	[136]
TRIM30α	RPT1	STING	K48	Degradation	[137]
		TAB2/3		Lysosomal degradation	[138]
TRIM31	RNF HCG1	MAVS	K63	Promotes MAVS signalosome assembly	[139]
TRIM32	TATIP BBS11 HT2A	Influenza PB1	K48	Degradation	[140]
		STING	K63	Activation	[141]
		TRIF	NA*	Targets TRIF for TAX1BP1-mediated autophagic degradation	[142]
TRIM33	TIF1γ	HIV integrase	K48	Degradation	[143]
TRIM35	HLS5 MAIR	TRAF3	K63	Activation	[144]
		IRF7	K48	Degradation	[145]
TRIM38	RNF15 RORET	RIG-I, MDA5	Sumoylation	Stabilization	[146]
		cGAS, STING	Sumoylation	Stabilization	[147]
		TRAF6	K48	Degradation	[148]
		NAP1	K48	Degradation	[149]
		TAB2	K48?	Degradation	[150]
		TRIF	K48	Degradation	[150,151]
TRIM39	RNF23 TFP	Cactin	NA*	Stabilizes Cactin, inhibiting NFκB and IRFs	[152]
TRIM40	RNF35	RIG-I, MDA5	K27, K48	Degradation	[153]
		NEMO	Neddylation	Inhibition	[154]
TRIM41	RINCK MGC1127	cGAS	Mono-Ub	Activation	[155]
TRIM44	DIPB AN3	MAVS	Interaction	Stabilization of MAVS by preventing its ubiquitination	[156]
TRIM45	RNF99	NFκB	E3 ligase activity not required	Inhibition of TNFα-mediated NFκB activation	[157]
TRIM56	RNF109	Influenza virus RNA	Inhibits vRNA synthesis		[158]
		cGAS	Mono-Ub	Activation	[159]
		STING	K63	Activation	[160]
		TRIF	Interaction	Activation	[161]
TRIM59	RNF104 TSBF1 MRF1 IFT80L	ECSIT	Interaction	Inhibition of TLR singling pathways to activate NFκB and IRFs	[162]
TRIM62	DEAR1	TRIF	NA*	Activation	[86]
TRIM65		MDA5	K63	Activation	
TRIM68	RNF137 SS56	TFG	various	Induces TFG lysosomal degradation	[163]

* NA: not assayed. Question marks (?) refer to “very likely but not experimentally revealed”.

## Data Availability

Not applicable.

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
