# Peer review of "TRIMming Type I Interferon-Mediated Innate Immune Response in Antiviral and Antitumor Defense"

_viruses, 2021, doi:10.3390/v13020279_

Round 1
Reviewer 1 Report
This is a well written and referenced review focusing on the role of the TRIM family of proteins in regulating the type 1 interferon pathway. This review will be a helpful resource for the rapidly growing field of TRIM protein biology.
There are only some very slight grammatical errors, which do not detract from the manuscript.
I have a couple of comments, the answers to which may help:
- There should be a greater explanation of why the focus on type 1 IFNs when there is significant similarity between type 1 and type 3 IFNs. Can the authors introduce a sentence outlining their reasoning or incorporate the - likely small - literature on the role of TRIMs in type 3 IFN immunity?
- Line 89 - not clear, except nfkb?
- Although the table of TRIMs is a great resource would their be room for a phylogenetic tree shoing the relationship between the TRIM family with regards to their functional biology?
- Would there be any worth in highlighting TRIM proteins for which there is no known function or indeed no known function in type 1 IFN pathway?
- Perspectives: Are there likely to be off target effects with CRISPR if TRIMs share sequence homology? How critical is this?
Author Response
REVIEWER 1:
- There should be a greater explanation of why the focus on type 1 IFNs when there is significant similarity between type 1 and type 3 IFNs. Can the authors introduce a sentence outlining their reasoning or incorporate the - likely small - literature on the role of TRIMs in type 3 IFN immunity?
RE: Type I IFN is our focus in this review, and also is line with our research scope and interest. We do not include other two types of IFNs. We added a sentence in the revision (Lines 72-74).
- Line 89 - not clear, except nfkb?
RE: Thanks for pickup. We have kept improving the manuscript after submission, including the correction of this error.
- Although the table of TRIMs is a great resource would their be room for a phylogenetic tree showing the relationship between the TRIM family with regards to their functional biology?
RE: This is a good point. But it can be found in several other reviews, which were mentioned and cited in this manuscript (Section 3; Refs. 49-58). We thus do not include this repeated information here, which is less important for this review.
- Would there be any worth in highlighting TRIM proteins for which there is no known function or indeed no known function in type 1 IFN pathway?
RE: A good point. We have added some information in "Perspectives" (Lines 482-483)
- Perspectives: Are there likely to be off target effects with CRISPR if TRIMs share sequence homology? How critical is this?
RE: We have clarified this with more information and a reference (Ref 231).
s

Reviewer 2 Report
The tripartite motif (TRIM) family of proteins includes at least 80 members in humans, with most having E3 ligase activity that can catalyze ubiquitination, ISGylation or SUMOylation. Several members of the TRIM superfamily are expressed in response to IFNs and are implicated in various biological processes associated with innate immunity and antiviral defense. In this review, the authors described the TRIM family protein domain alignment and summarized the effects of TRIMs in regulating PRR signaling and Jak-STAT IFN-I signaling.
There are many errors in the cited references in the text and in table 1. the authors must verify that each reference is correctly cited in the article
1-line 304 : the following sentence is not correct
« TRIM19IV stabilizes p-IRF3 by targeting Pin1 for degradation »
and should be modified by « TRIM19IV stabilizes p-IRF3 by recruiting Pin1 within PML nuclear bodies »
2-Line 317 : the reference 149 is not well cited
« TRIM19/PML, as a transcriptional repressor, suppresses NFκB-mediated gene tran- scription via its C-terminus independently of its E3 ligase activity [149] ».
In the reference 149 it has been shown that « PML promotes TNFα-induced transcriptional responses by promoting NF-κB activity ».
Therefore, the authors should either add the correct reference or modify the sentence.
3- This reference “Chang, T.-H.; Yoshimi, R.; Ozato, K. TRIM12c, a Mouse Homolog of TRIM5, Is a Ubiquitin Ligase That Stimulates Type I IFN 556 and NF-κB Pathways along with TNFR-Associated Factor 6. J Immunol 2015, 195, 5367-5379 » is cited in Table 1 as [57] and also as [104]. This is unacceptable. The authors must verify all the cited references.
4- Line 357: The following sentence should be modified :
“The nuclear protein TRIM19/PML mediates STAT1 and -2 transcriptional induction [156,157] ».
In the reference [156] the authors showed that « PML facilitates the transcription of STAT1 and the stable expression of STAT2 protein ».
In addition, the reference [157] is not correctly cited and should be deleted. The reference [157] could be cited in the section « Viral strategies to subvert TRIM-mediated IFN-I regulatory mechanisms».
5- Line369
Ubiquitination-like modifications, such as sumoylation and ISGylation, are involved in IFN-I-mediated defense mechanisms [29,163]. The references on the effect of SUMOylation and ISGylation of IFN responses are missing.
These three references should be added
a- Hannoun Z et al. The implication of SUMO in intrinsic and innate immunity
Cytokine Growth Factor Rev 2016 Jun;29:3-16.
b-Harty RN, Pitha PM et al. Antiviral activity of innate immune protein ISG15. J Innate Immun. 2009;1(5):397-404.
c- Freitas BT et al. How ISG15 combats viral infection.Virus Research. Sept, 2020, Vol. 286.
5- Figure 2 is not indicated in the body of the text
6- Line 371
« TRIM25 functions as an ISG15 E3 ligase that mediates ISGylation [165] ». After the reference [165] the authors could mention that TRIM25 has been recently reported to be required for the stability of several ISG products (El Asmi F et al. Cytokine 2020 May;129:155025).
7- Line 374
« The tumor suppressor p53 is an ISG inducible by IFN-Is »
The reference showing that p53 is an ISG is missing. Add the following reference
Takaoka, Akinori et al. Nature. July 31, 2003, Vol. 424 Issue 6948, 516, 8.
8-line 407 in « Viral strategies to subvert TRIM-mediated IFN-I regulatory mechanisms »
The authors should also add :
- EMCV 3C protease promotes PML degradation
El Mchichi B, Regad T et al. (J Virol. 2010 Nov; 84(22): 11634–11645).
- The rabies viral protein P also causes the redistribution of PML into the cytoplasm where the proteins colocalize
(Blondel D et al. 2002. Oncogene 21:7957-7970).
9- In the legend to Figure 1, the significance of ISRE3/7 and ISRE7 should be precised
10- In table 1
The following data should be added in table 1
TRIM19IV «interacts with 3D polymerase » « inhibits EMCV replication »
Reference :Mohamed Ali Maroui et al., Promyelocytic leukemia isoform IV confers resistance to encephalomyocarditis virus via the sequestration of 3D polymerase in nuclear bodies. J. Virol. 2011;85(24):13164-73.
TRIM19IV « sequesters virion capsids » « inhibits VZV replication »
Reference :Reichelt et al.Entrapment of viral capsids in nuclear PML cages is an intrinsic antiviral host defense against varicella-zoster virus. PLoS Pathog2011,7:e1001266.
TRIM22 « ubiquitinates EMCV 3C protease » « inhibits EMCV replication »
Reference :Eldin P et al., TRIM22 E3 ubiquitin ligase activity is required to mediate antiviral activity against encephalomyocarditis virus. J Gen Virol. 2009;90:536-545.
TRIM79α « degradation viral polymerase » « resricts viral replication »
R Travis Taylor et al., TRIM79α, an interferon-stimulated gene product, restricts tick-borne encephalitis virus replication by degrading the viral RNA polymerase. Cell Host Microbe . 2011 Sep 15;10(3):185-96.
11- In table 1
In the top of table 1 add ubiquitination to “conjugation type”
Many references are not correctly cited such as 128, 151, 152, 153, 167 etc…..
Take of the ? and replace by NA*
Author Response
REVIEWER 2:
There are many errors in the cited references in the text and in table 1. the authors must verify that each reference is correctly cited in the article
RE: We have kept improving the writing after submission. We make sure all references are correct, including the merge of duplicate references. Some errors in the numbering of references may be caused by the reorganization of the manuscript compiled by the editorial assistance after submission to comply with the journal VIRUSES template (Table 1 was moved to the main body but references’ order may not be automatically resorted). All numbers have been automatically re-sorted by Endnote in the revision.
1-line 304 : the following sentence is not correct. « TRIM19IV stabilizes p-IRF3 by targeting Pin1 for degradation » and should be modified by « TRIM19IV stabilizes p-IRF3 by recruiting Pin1 within PML nuclear bodies »
RE: Thanks for picking up this mistake, which is corrected in the revision.
2-Line 317 : the reference 149 is not well cited. « TRIM19/PML, as a transcriptional repressor, suppresses NFκB-mediated gene transcription via its C-terminus independently of its E3 ligase activity [149] ». In the reference 149 it has been shown that « PML promotes TNFα-induced transcriptional responses by promoting NF-κB activity ». Therefore, the authors should either add the correct reference or modify the sentence.
RE: Thanks. We include both controversial studies in the revision. Lines 327-331.
3- This reference “Chang, T.-H.; Yoshimi, R.; Ozato, K. TRIM12c, a Mouse Homolog of TRIM5, Is a Ubiquitin Ligase That Stimulates Type I IFN 556 and NF-κB Pathways along with TNFR-Associated Factor 6. J Immunol 2015, 195, 5367-5379 » is cited in Table 1 as [57] and also as [104]. This is unacceptable. The authors must verify all the cited references.
RE: Fixed. See response above.
4- Line 357: The following sentence should be modified: “The nuclear protein TRIM19/PML mediates STAT1 and -2 transcriptional induction [156,157] ». In the reference [156] the authors showed that « PML facilitates the transcription of STAT1 and the stable expression of STAT2 protein ». In addition, the reference [157] is not correctly cited and should be deleted. The reference [157] could be cited in the section « Viral strategies to subvert TRIM-mediated IFN-I regulatory mechanisms».
RE: Thanks for this critique. Fixed by following the suggestions
5- Line369
Ubiquitination-like modifications, such as sumoylation and ISGylation, are involved in IFN-I-mediated defense mechanisms [29,163]. The references on the effect of SUMOylation and ISGylation of IFN responses are missing. These three references should be added......
RE: Added as suggested.
5- Figure 2 is not indicated in the body of the text
RE: It is in Section 3
6- Line 371
« TRIM25 functions as an ISG15 E3 ligase that mediates ISGylation [165] ». After the reference [165] the authors could mention that TRIM25 has been recently reported to be required for the stability of several ISG products (El Asmi F et al. Cytokine 2020 May;129:155025).
RE: Thanks. Added as suggested. Lines 388-389.
7- Line 374
« The tumor suppressor p53 is an ISG inducible by IFN-Is ». The reference showing that p53 is an ISG is missing. Add the following reference. Takaoka, Akinori et al. Nature. July 31, 2003, Vol. 424 Issue 6948, 516, 8.
RE: Added as suggested. Line 393.
8-line 407 in « Viral strategies to subvert TRIM-mediated IFN-I regulatory mechanisms » The authors should also add :
- EMCV 3C protease promotes PML degradation El Mchichi B, Regad T et al. (J Virol. 2010 Nov; 84(22): 11634–11645).
- The rabies viral protein P also causes the redistribution of PML into the cytoplasm where the proteins colocalize (Blondel D et al. 2002. Oncogene 21:7957-7970).
RE: Added as suggested. Lines 437-439
9- In the legend to Figure 1, the significance of ISRE3/7 and ISRE7 should be precised
RE: Fixed.
10- In table 1. The following data should be added in table 1………
RE: Thanks for these suggestions. As we mentioned in the beginning of Section 4 (Lines 147-148), table 1 only includes some examples. There are too many studies on how TRIMs regulate viral proteins, and we cannot include all of them in this review. We cite two reviews here (Refs 54, 58). Lines 148-149. We also added a new reference, as suggested by the Editor (Lines, 149-152. Ref 66)
11- In table 1. In the top of table 1 add ubiquitination to “conjugation type”
RE: Added as suggested.
Many references are not correctly cited such as 128, 151, 152, 153, 167 etc…..
RE: See response above. We have improved the writing after submission, including the correction of some references.
Take of the ? and replace by NA*
RE: "?" in Table 1 means "Very likely but not experimentally revealed". NA means it is not known what kind of conjugation type. We try to distinguish these two possibilities so we still keep ?. We now clarify this difference in table legend
Editor:
- Reference 159 is duplicated (159 and 160 seem to be the same reference.
Please choose one).
RE: See response above, we have fixed several duplicates after submission, including those found by reviewers
- I suggest to add discussion/information on additional TRIMs that subvert the IFN-I response and /or are hijacked by viruses. Discussion on Nipah virus antagonism of TRIM6-mediated IFN response (see PMID: 27622505).
RE: Revised as suggested. See lines 465-469.
- And the information on TRIM7 could be expanded to include
its role in ubiquitination of Zika virus envelope protein and enhancing virus replication (see PMID: 32641828).
RE: Revised as suggested. See Table 1, and Lines 149-151.

Round 2
Reviewer 2 Report
The authors took into account all remarks and made the corrections in the article and the references